# Artemisinin exposure at the ring or trophozoite stage impacts *Plasmodium falciparum* sexual conversion differently

**Harvie P Portugaliza**[1,2,3], **Shinya Miyazaki**[4], **Fiona JA Geurten**[4], **Christopher Pell**[3,5], **Anna Rosanas-Urgell**[2], **Chris J Janse**[4], **Alfred Cortés**[1,6]*

[1]ISGlobal, Hospital Clinic – Universitat de Barcelona, Barcelona, Spain; [2]Department of Biomedical Sciences, Institute of Tropical Medicine, Antwerp, Belgium; [3]Department of Global Health, Amsterdam University Medical Centers, location Academic Medical Center, University of Amsterdam, Amsterdam, Netherlands; [4]Department of Parasitology, Leiden University Medical Center, Leiden, Netherlands; [5]Amsterdam Institute for Global Health and Development (AIGHD), Amsterdam, Netherlands; [6]ICREA, Barcelona, Spain

**Abstract** Malaria transmission is dependent on the formation of gametocytes in the human blood. The sexual conversion rate, the proportion of asexual parasites that convert into gametocytes at each multiplication cycle, is variable and reflects the relative parasite investment between transmission and maintaining the infection. The impact of environmental factors such as drugs on sexual conversion rates is not well understood. We developed a robust assay using gametocyte-reporter parasite lines to accurately measure the impact of drugs on sexual conversion rates, independently from their gametocytocidal activity. We found that exposure to subcurative doses of the frontline antimalarial drug dihydroartemisinin (DHA) at the trophozoite stage resulted in a ∼ fourfold increase in sexual conversion. In contrast, no increase was observed when ring stages were exposed or in cultures in which sexual conversion was stimulated by choline depletion. Our results reveal a complex relationship between antimalarial drugs and sexual conversion, with potential public health implications.

**\*For correspondence:**
alfred.cortes@isglobal.org

**Competing interests:** The authors declare that no competing interests exist.

## Introduction

*Plasmodium falciparum* is responsible for the most severe forms of human malaria. Repeated rounds of its ~48 hr intraerythrocytic asexual replication cycle result in an exponential increase in parasite numbers and are responsible for all clinical symptoms of malaria. At each round of replication, a small subset of parasites commits to differentiation into non-replicative sexual forms termed gametocytes, which are the only form transmissible to a mosquito vector. Sexual commitment is marked by epigenetic activation of the expression of the master regulator PfAP2-G, a transcription factor of the ApiAP2 family (*Josling et al., 2020*; *Kafsack et al., 2014*; *Llorà-Batlle et al., 2020*; *Poran et al., 2017*). This is followed by sexual conversion, which according to our recently proposed definitions (*Bancells et al., 2019*) is marked by the expression of gametocyte-specific proteins absent from any replicating blood stages. After sexual conversion, parasites at the sexual ring stage develop through gametocyte stages I to V in a maturation process that lasts for ~10 days (*Josling et al., 2018*; *Ngotho et al., 2019*). While immature gametocytes are sequestered in organs such as the bone marrow (*Venugopal et al., 2020*), mature gametocytes (stage V) are released into the circulation, where they are infectious to mosquitoes for several days or even weeks (*Cao et al., 2019*). To eliminate malaria, which the World Health Organization has adopted as a global goal, it is necessary to

block transmission, as well as killing asexual parasites to cure patients (*World Health Organization, 2017*).

To secure within-host survival and between-host transmission, the proportion of parasites that convert into sexual forms at each replicative cycle, termed sexual conversion rate, is variable and tightly regulated. In human infections, gametocyte densities are always much lower than asexual parasite densities, and basal *P. falciparum* sexual conversion rates in vivo are estimated to be ~1% (*Cao et al., 2019*; *Eichner et al., 2001*). This reveals a reproductive restraint for which multiple alternative hypotheses have been proposed (*McKenzie and Bossert, 1998*; *Mideo and Day, 2008*; *Taylor and Read, 1997*). Whatever the reason for the low levels of sexual conversion, multiple observations suggest that malaria parasites can respond to changes in the conditions of their environment by adjusting the trade-off between transmission and within-host survival. From an evolutionary perspective, the ability to adjust sexual conversion rates depending on the host conditions would be clearly advantageous for the parasite (*Carter et al., 2013*; *Schneider et al., 2018*). In *P. falciparum*, several conditions have been shown to increase sexual conversion rates, and sexual conversion is viewed by some as a response to stress, although this remains controversial (*Baker, 2010*; *Bousema and Drakeley, 2011*; *Dyer and Day, 2000*; *Josling et al., 2018*). The most commonly used method to enhance sexual conversion and obtain large numbers of gametocytes in vitro relies on overgrowing blood-stage cultures (the 'crash method') (*Delves et al., 2016*) and/or maintaining the cultures with parasite-conditioned (spent) medium (*Brancucci et al., 2015*; *Fivelman et al., 2007*). Recent research has established that depletion of the human serum lipid lysophosphatidylcholine (LysoPC) underlies the stimulation of sexual conversion by high asexual parasitemia or spent medium, providing the first mechanistic insight into how environmental conditions can influence the rate of sexual conversion (*Brancucci et al., 2018*; *Brancucci et al., 2017*). Low plasma LysoPC levels were also associated with increased sexual conversion rates in human infections (*Usui et al., 2019*). Depletion of LysoPC or choline, a downstream metabolite in the same metabolic pathway, has now been used by several groups to stimulate sexual conversion under culture conditions (*Brancucci et al., 2017*; *Filarsky et al., 2018*; *Portugaliza et al., 2019*).

Artemisinin and its derivatives (collectively referred to as ARTs) are potent antimalarial drugs that rapidly kill asexual parasites. After activation by cleavage of their endoperoxide bond by hemoglobin degradation products, ARTs produce reactive oxygen species and free radicals that result in widespread damage in parasite proteins and lipids. However, because ARTs have a very short elimination half-life in the human circulation (~1–3 hr), their application as monotherapy was discontinued to avoid infection recrudescence and development of drug resistance. Artemisinin-based combination therapies (ACTs), consisting of ART and a long-acting partner drug, are the current frontline treatment for uncomplicated as well as severe malaria cases (*Blasco et al., 2017*; *de Vries and Dien, 1996*; *Haldar et al., 2018*; *Talman et al., 2019*). Resistance to ARTs has emerged in South-East Asia in the form of delayed parasite clearance (*Dondorp et al., 2009*). ART resistance is associated with mutations in the PfKelch13 protein (*Ariey et al., 2014*) that prevent hemoglobin degradation in early ring-stage parasites. This in turn prevents ART activation, resulting in resistance of early rings to the drug (*Birnbaum et al., 2020*; *Yang et al., 2019*). Nowadays, ART resistance is frequently accompanied by simultaneous resistance to partner drugs such as mefloquine, piperaquine, or amodiaquine, resulting in high rates of treatment failure and limiting treatment options (*Mairet-Khedim et al., 2020*; *Phyo et al., 2016*; *van der Pluijm et al., 2019*).

Treatment with antimalarial drugs such as chloroquine (CQ) or sulfadoxine-pyrimethamine is usually associated with increased gametocytemia (density of gametocytes in the blood) on the days following drug administration, whereas treatment with ACTs results in reduced gametocytemia and transmission to mosquitoes (*Ippolito et al., 2017*; *Okell et al., 2008*; *Price et al., 1996*; *Sawa et al., 2013*; *von Seidlein et al., 2001*; *WWARN Gametocyte Study Group, 2016*). Despite the efficacy of ACTs in reducing gametocytemia, successfully treated patients can remain infectious for several days and contribute to transmission (*Bousema et al., 2006*; *Bousema et al., 2010*; *Karl et al., 2015*; *Targett et al., 2001*). The higher capacity of ACTs to reduce gametocytemia compared to other drugs is attributable to several factors: (i) faster killing of asexual parasites, which prevents the formation of new gametocytes; (ii) more efficient killing of immature gametocytes; (iii) partial clearance of mature gametocytes, which are insensitive to most other clinically relevant drugs (*Adjalley et al., 2011*; *Chotivanich et al., 2006*; *Plouffe et al., 2016*).

Although it has been proposed that the increase of gametocytemia observed after treatment with some drugs may reflect stimulation of sexual conversion, there is no direct linear relationship between conversion rates and the prevalence and density of circulating gametocytes (*Carter et al., 2013*; *Koepfli and Yan, 2018*; *Reece and Schneider, 2018*). The dynamics of circulating gametocyte densities after treatment can be explained without invoking an adjustment of sexual conversion rates: first, gametocytes are sequestered away from the circulation until ~10 days after sexual conversion, implying that the peaks of gametocytemia observed after treatment with some drugs (within less than 10 days) may reflect the dynamics of asexual parasite growth before treatment, rather than post-treatment changes in sexual conversion. Second, the effects of the drugs on sexual conversion rates in human infections cannot be disentangled from other drug-mediated actions such as the release of sequestered parasites or gametocyte clearance (*Babiker et al., 2008*; *Bousema and Drakeley, 2011*; *Butcher, 1997*; *Koepfli and Yan, 2018*).

To directly address the effect of drug treatment on sexual conversion, a small number of studies have used *P. falciparum* in vitro cultures, yielding inconsistent results. While some studies reported increased sexual conversion upon exposure to specific doses of drugs such as CQ or ART (*Buckling et al., 1999b*; *Peatey et al., 2009*; *Rajapandi, 2019*), others did not observe this effect with ART (*Brancucci et al., 2015*), or reported reduced sexual conversion upon exposure to low doses of CQ or pyrimethamine (*Reece et al., 2010*). Although the discrepancies may reflect methodological differences between these studies and limited accuracy in determining sexual conversion rates, the divergent conclusions also suggest a complex scenario in which conditions such as the specific drug used, the parasite stage at the time of exposure, and drug concentration may determine the effect of treatment on sexual conversion.

Given the widespread use of ACTs for malaria treatment and in mass drug administration campaigns aimed at malaria elimination, understanding the impact of ARTs on sexual conversion is an urgent research priority. Here we developed a robust assay based on recently described gametocyte-reporter parasite lines (*Portugaliza et al., 2019*) to accurately measure the impact of drugs on sexual conversion rates, independently from their gametocytocidal activity. Using this assay, we tested the effect of exposing parasites to dihydroartemisinin (DHA, the active metabolite of all ARTs) at different stages and under different metabolic conditions, to provide an accurate and comprehensive description of the direct effect of this drug on sexual conversion rates. We also tested the effect of another drug, CQ, and a different type of stress, heat shock, on sexual conversion rates.

## Results

### Exposure to DHA at the trophozoite stage enhances sexual conversion

To examine the effect of ARTs on *P. falciparum* sexual conversion, we administered a 3 hr pulse of DHA to synchronous cultures of the *NF54-gexp02-Tom* reporter line. This parasite line expresses the fluorescent reporter tdTomato under the control of the promoter of the sexual stage-specific gene *gexp02* (PF3D7_1102500), which allows accurate flow cytometry-based detection of very early gametocytes within a few hours after sexual conversion (*Portugaliza et al., 2019*). The short drug pulse mimics the short plasma half-life of ARTs (*de Vries and Dien, 1996*). Cultures were regularly maintained in choline-containing culture medium (Albumax-based medium with a supplement of choline) to mimic the repression of sexual conversion by healthy human serum, and choline was either maintained or removed during the experiment to repress or stimulate sexual conversion (*Brancucci et al., 2017*; *Filarsky et al., 2018*). The DHA pulse was administered at the trophozoite (*Figure 1A*) or the ring (see below) stage, using subcurative DHA concentrations (5 and 10 nM) that in trophozoites resulted in a reduction of growth of <40% (*Figure 1B*). The sexual conversion rate was calculated as the proportion of parasites that developed into gametocytes at the cycle after exposure (i.e., after reinvasion) (*Figure 1—figure supplement 1A*).

In cultures supplemented with choline, the sexual conversion rate increased from <10% in control cultures to ~40% in cultures exposed to a 5 or 10 nM DHA pulse at the trophozoite stage (*Figure 1C–D*). Importantly, total gametocytemia (determined at the next multiplication cycle after drug exposure, i.e., early gametocytes) was also clearly higher in DHA-exposed cultures than in control cultures (*Figure 1E*). This result indicates that the increase in the sexual conversion rate is not

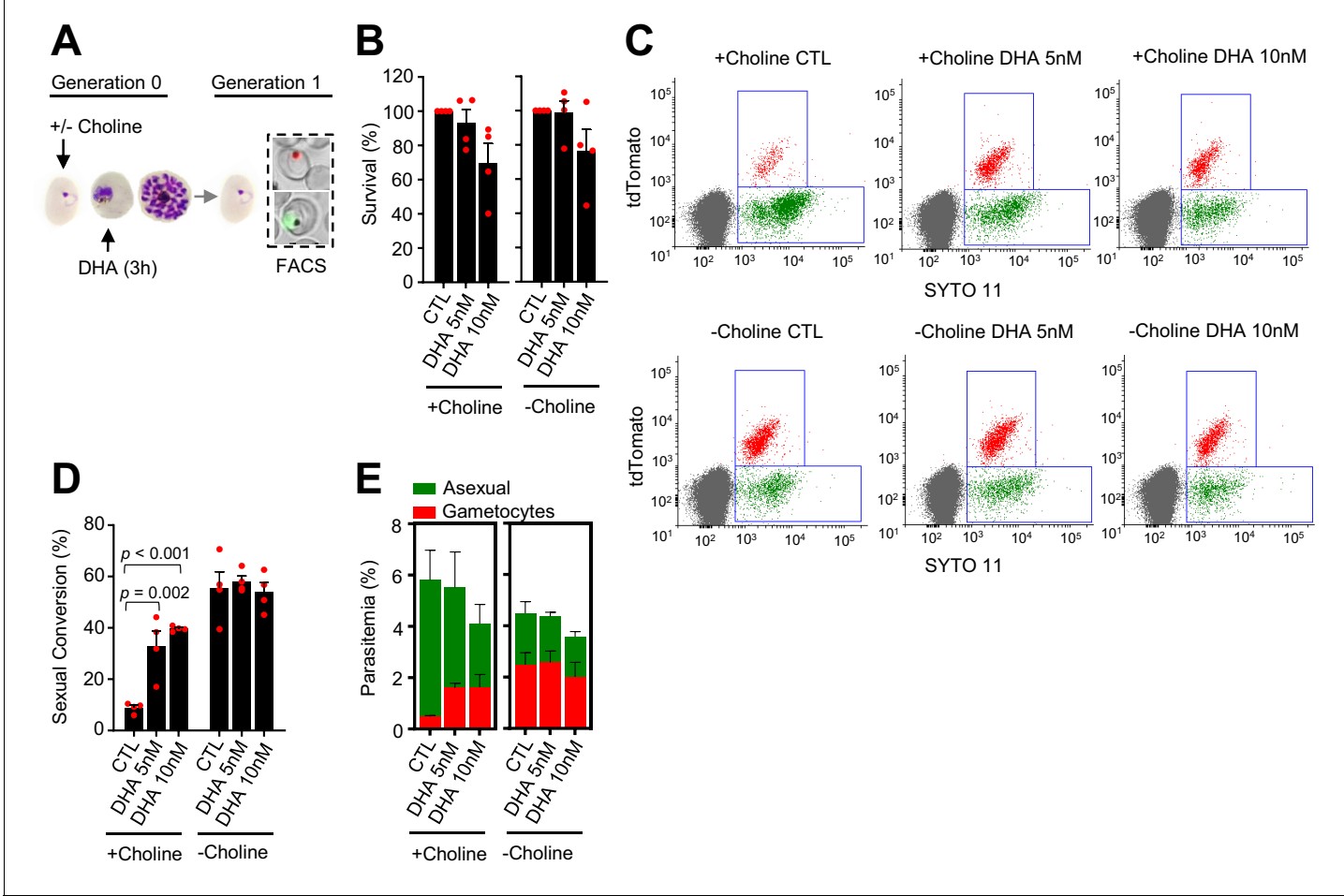

**Figure 1.** Effect of a dihydroartemisinin (DHA) pulse at the trophozoite stage on sexual conversion. (**A**) Schematic representation of the assay. Tightly synchronized cultures of the *NF54-gexp02-Tom* line under non-inducing (+ choline) or inducing (– choline) conditions were exposed to a 3 hr DHA pulse at subcurative doses at the trophozoite stage (25–30 hpi). Sexual conversion was measured by flow cytometry (FACS) after reinvasion (~30–35 hpi of the next multiplication cycle). (**B**) Survival rate of cultures exposed to the different drug doses, using total parasitemia values (asexual + sexual parasites). For each choline condition, values are presented relative to the parasitemia in the control cultures (no drug). (**C**) Representative SYTO 11 (stains parasite DNA) versus TdTomato (marks gametocytes) flow cytometry plots. (**D**) Sexual conversion rate determined by flow cytometry. The p-value is indicated only for treatment versus control (no drug) significant differences (p<0.05). (**E**) Distribution of absolute parasitemia of asexual and sexual parasites (from the same flow cytometry measurements as in panel D). In all panels, data are presented as the average and s.e.m. of four independent biological replicates.

The online version of this article includes the following figure supplement(s) for figure 1:

**Figure supplement 1.** Schematic representation of the assays to determine sexual conversion rates.

**Figure supplement 2.** Effect of a dihydroartemisinin (DHA) pulse at the trophozoite stage on sexual conversion, determined using MitoTracker to identify viable parasites.

**Figure supplement 3.** Effect of a dihydroartemisinin (DHA) pulse at the trophozoite stage on sexual conversion in the *E5-gexp02-Tom* line.

**Figure supplement 4.** Effect of a dihydroartemisinin (DHA) pulse at the trophozoite stage in the NF54-10.3-Tom line on sexual conversion, determined by three different methods.

**Figure supplement 5.** Effect of a low concentration dihydroartemisinin (DHA) pulse at the trophozoite stage on sexual conversion.

**Figure supplement 6.** Effect on sexual conversion of a dihydroartemisinin (DHA) pulse at 5–30 nM concentrations during the trophozoite stage.

**Figure supplement 7.** Flow cytometry set-up for the identification of viable parasites in the 3D7-A parasite line using MitoTracker.

**Figure supplement 8.** Flow cytometry set-up for the identification of viable parasites in the *NF54-gexp02-Tom* line using MitoTracker.

only attributable to the lower number of asexual parasites after drug treatment, but also to a net increase in the number of gametocytes produced. By contrast, in cultures in which sexual conversion was already stimulated by choline depletion, DHA treatment did not result in a further increase in the sexual conversion rate or in the absolute number of gametocytes (*Figure 1C–E*). Similar results

were obtained in experiments in which sexual conversion rates were calculated based only on viable parasites as identified by a marker of active mitochondria (*Figure 1—figure supplement 2*), using an analogous reporter line generated in the 3D7-E5 genetic background that has lower levels of basal sexual conversion than NF54 (*E5-gexp02-Tom* line, *Figure 1—figure supplement 3*; *Portugaliza et al., 2019*), and using a transgenic line with the fluorescent reporter under the control of the *etramp10.3* (PF3D7_1016900) gametocyte-specific promoter (*NF54-10.3-Tom* line) (*Portugaliza et al., 2019*). Using this latter parasite line, we measured sexual conversion rates by flow cytometry, by immunofluorescence assay (IFA) detecting the Pfs16 (PF3D7_0406200) early gametocyte marker and by light microscopy analysis of Giemsa-stained blood smears (*Figure 1—figure supplement 1B*). All approaches yielded similar results and confirmed enhanced sexual conversion after exposure of trophozoites to subcurative doses of DHA (*Figure 1—figure supplement 4*).

In an additional set of experiments, we tested the impact of a DHA pulse at lower concentrations ($\leq$5 nM) on sexual conversion (*Figure 1—figure supplement 5*). A 2 nM DHA pulse did not have a measurable effect on parasite growth (*Figure 1—figure supplement 5B*), but in choline-supplemented cultures it consistently resulted in a > twofold increase in the sexual conversion rate and the total number of gametocytes (although not statistically significant; *Figure 1—figure supplement 5C–D*). Drug doses < 2 nM did not have a detectable effect on sexual conversion.

We also tested the impact on sexual conversion of higher DHA doses up to 30 nM, a concentration that kills ~90% of the parasites (*Figure 1—figure supplement 6A–B*). In choline-supplemented cultures, both sexual conversion rates and total gametocytemia were clearly enhanced upon exposure to DHA concentrations up to 15 nM, but the increase was lower upon exposure to higher concentrations. In choline-depleted cultures, increasing DHA doses resulted in a progressive reduction of sexual conversion (*Figure 1—figure supplement 6C–D*). However, it is important to note that the determination of sexual conversion rates is less accurate when the majority of the parasites are killed by the drug. Thus, given that maximum induction was observed at 10 nM and the difficulties to estimate sexual conversion accurately in experiments with higher drug doses, we used 5 and 10 nM DHA pulses for the majority of experiments described in the next sections.

Gametocytes of the *NF54-gexp02-Tom* line produced in cultures treated with 5 nM DHA at the trophozoite stage (in the presence of choline) matured through stages I to V without any apparent morphological alteration. Furthermore, mature, stage V gametocytes were able to exflagellate after activation and to infect mosquitoes productively (*Figure 2*). As expected from the increase in the number of early gametocytes in DHA-treated cultures, the gametocytemia 10–13 d later (mature gametocytes) and the number of exflagellation centers were also higher in DHA-treated cultures compared to control cultures (no DHA). Likewise, when using these cultures to infect mosquitoes, the number of oocysts and sporozoites per mosquito was higher in mosquitoes fed with DHA-treated cultures than in mosquitoes fed with control cultures. The magnitude of the increase was similar to that observed in mosquitoes fed with cultures in which sexual conversion was stimulated by choline depletion (*Figure 2B*). While only differences in the number of exflagellation centers and oocysts/mosquito were statistically significant (p<0.05) between DHA-treated and control cultures (which is likely attributable to the intrinsic variability of mosquito feeding experiments), a clear increase was observed in all independent biological replicates for all parameters tested.

## DHA exposure at the ring stage does not enhance sexual conversion

A DHA pulse (5 or 10 nM) at the early ring stage that reduced growth by <25% (*Figure 3A–B*) did not enhance sexual conversion. Instead, it resulted in a reduction of sexual conversion and gametocytemia, both in choline-supplemented and choline-depleted *NF54-gexp02-Tom* cultures (*Figure 3C–E*; *Figure 3—figure supplement 1*). This unexpected result was confirmed using the *NF54-10.3-Tom* reporter line and the different methods described above to assess sexual conversion (*Figure 3—figure supplement 2*). In an additional set of experiments using *NF54-gexp02-Tom* cultures, we tested higher concentrations of the drug, and again observed a decrease in sexual conversion rates that in choline-depleted cultures was more marked with higher concentrations of the drug (*Figure 3—figure supplement 3*). In these experiments, we determined sexual conversion by measuring expression of the gametocyte reporter at 24 and 48 hr post invasion (hpi), and observed no difference between sexual conversion determined at the two different time points. This result excludes the possibility that the lower conversion rates observed in cultures exposed to DHA at the

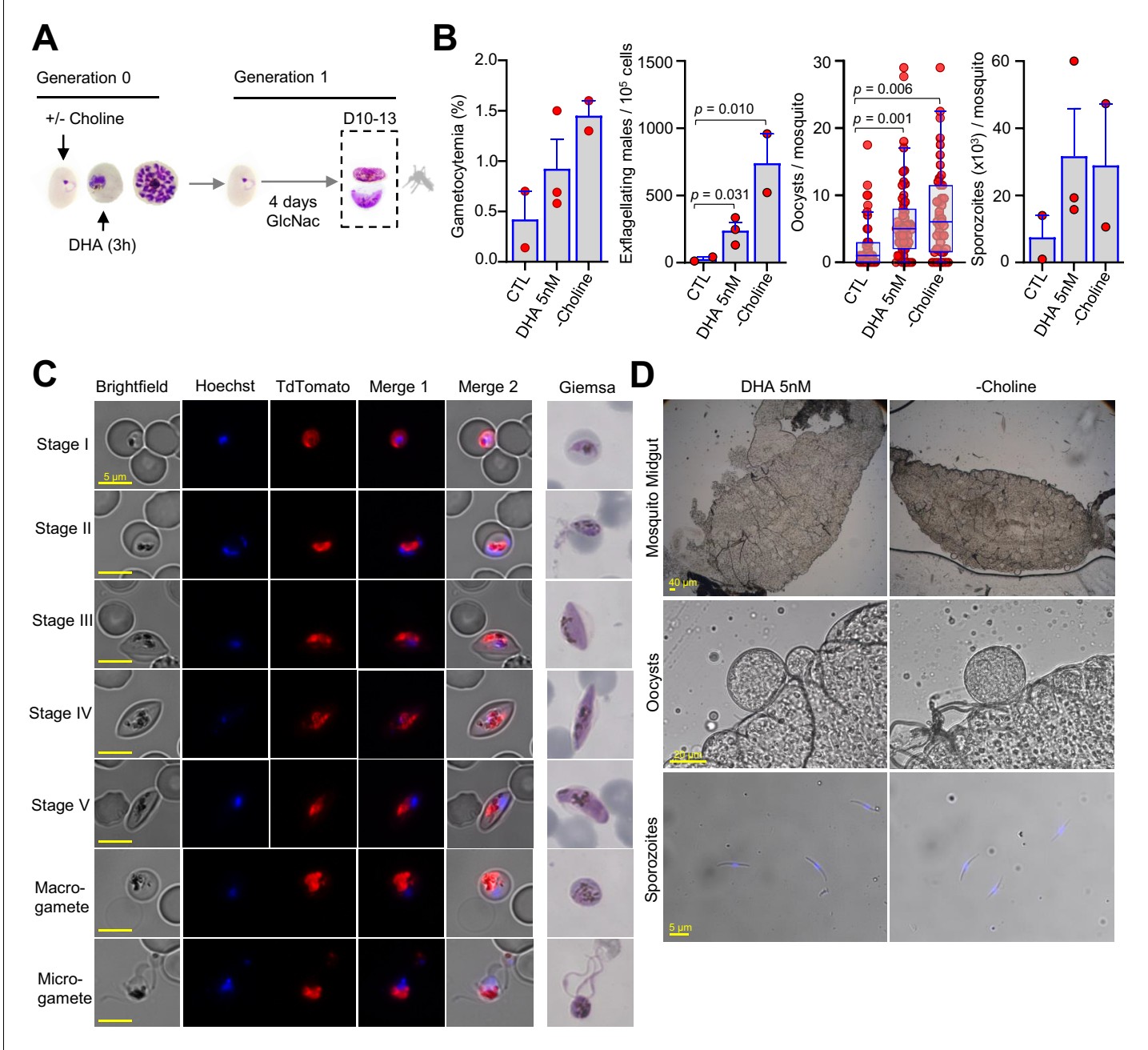

**Figure 2.** Mosquito infection by gametocytes from cultures exposed to dihydroartemisinin (DHA). (**A**) Schematic representation of the assay. Sorbitol-synchronized cultures of the *NF54-gexp02-Tom* line were maintained under control non-inducing conditions (+ choline, CTL), exposed to a 3 hr 5 nM DHA pulse (in the presence of choline) at the trophozoite stage, or maintained in the absence of choline (– choline, used as a positive control for a gametocyte-inducing condition). On the first day of Generation 1, N-acetylglucosamine (GlcNAc) was added and maintained for 4 d to eliminate asexual parasites and obtain pure gametocyte cultures. The different cultures were used to infect *Anopheles* mosquitoes by standard membrane feeding. (**B**) Gametocytemia at the time of mosquito infection (10–13 d after DHA treatment), exflagellation levels (after 10 min of activation with fetal calf serum), number of oocysts/mosquito (n = 53 for CTL, n = 103 for DHA 5 nM, and n = 79 for – choline, data for all individually dissected mosquitoes from all replicates is shown) and average number of sporozoites/mosquito in each independent biological replicate (obtained from pooled dissections; in total, n = 65 for CTL; n = 111 for DHA; n = 123 for – choline). Results are from three independent biological replicates, but in one experiment the CTL culture was lost and in another one the – choline control was not included. Data are presented as the average and s.e.m. of the independent biological replicates, except for oocysts/mosquito results that are presented as standard box and whisker plots. The p-value is indicated only for significant differences (p<0.05) between conditions. (**C**) Representative images of gametocytes at different stages and activated gametes from DHA-treated cultures, showing no apparent abnormality. Images from live cell fluorescence analysis (Hoechst stains nuclei; TdTomato is expressed under the

*Figure 2 continued on next page*

*Figure 2 continued*

control of the *gexp02* promoter) and Giemsa-stained smears are shown. (**D**) Representative images of mosquito midguts (transparent, circular structures are oocysts), oocysts, and sporozoites from DHA-treated and – choline cultures, showing no apparent abnormality.

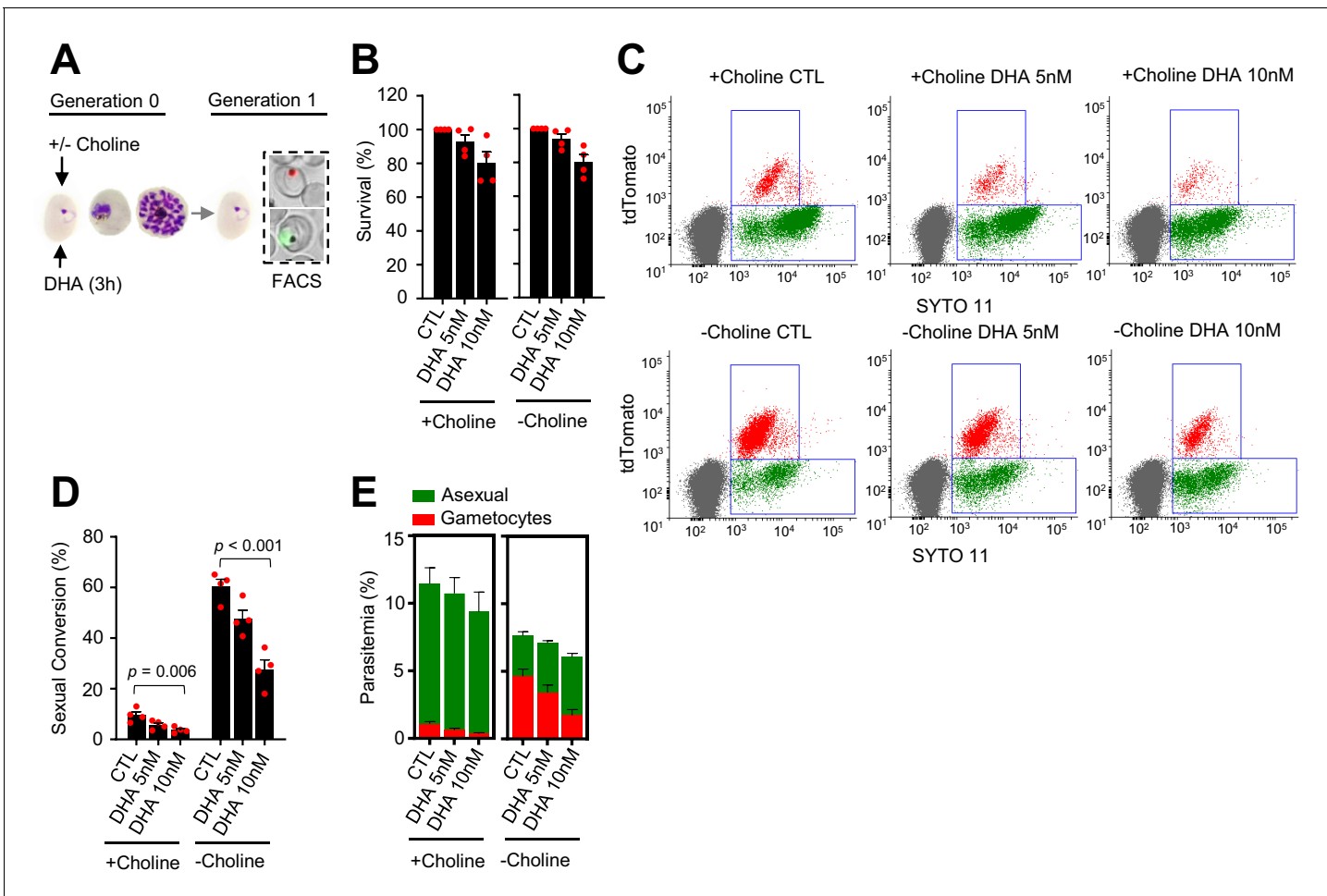

**Figure 3.** Effect of a dihydroartemisinin (DHA) pulse at the ring stage on sexual conversion. (**A**) Schematic representation of the assay. Tightly synchronized cultures of the *NF54-gexp02-Tom* line under non-inducing (+ choline) or inducing (– choline) conditions were exposed to a 3 hr DHA pulse at subcurative doses at the early ring stage (0–10 hpi). Sexual conversion was measured by flow cytometry (FACS) after reinvasion (~30–40 hpi of the next multiplication cycle). (**B**) Survival rate of cultures exposed to the different drug doses, using total parasitemia values (asexual + sexual parasites). For each choline condition, values are presented relative to the parasitemia in the control cultures (no drug). (**C**) Representative SYTO 11 (stains parasite DNA) versus TdTomato (marks gametocytes) flow cytometry plots. (**D**) Sexual conversion rate determined by flow cytometry. The p-value is indicated only for treatment versus control (no drug) significant differences (p<0.05). (**E**) Distribution of absolute parasitemia of asexual and sexual parasites (from the same flow cytometry measurements as in panel D). In all panels, data are presented as the average and s.e.m. of four independent biological replicates.

The online version of this article includes the following figure supplement(s) for figure 3:

**Figure supplement 1.** Effect of a dihydroartemisinin (DHA) pulse at the ring stage on sexual conversion, determined using MitoTracker to identify viable parasites.

**Figure supplement 2.** Effect on sexual conversion of a dihydroartemisinin (DHA) pulse at the ring stage in the *NF54-10.3-Tom* line, determined by three different methods.

**Figure supplement 3.** Effect on sexual conversion of a dihydroartemisinin (DHA) pulse at 10–40 nM concentrations at the ring stage.

ring stage are attributable to a DHA-induced delay in sexual conversion or gametocyte development (*Figure 3—figure supplement 3*).

To explore the possibility that DHA exposure at the early ring stage may stimulate immediate sexual conversion via the same cycle conversion (SCC) pathway (*Bancells et al., 2019*), rather than by the canonical next cycle conversion (NCC) pathway, we assessed the effect of DHA exposure at the ring stage on the level of gametocytes produced within the same cycle of exposure (*Figure 4A*). We observed no apparent differences in sexual conversion rates via the SCC route between DHA-exposed cultures and their controls (*Figure 4B–C*; *Figure 4—figure supplement 1*). Similar results were obtained using the *NF54-10.3-Tom* reporter line and flow cytometry or light microscopy analysis of Giemsa-stained smears to measure sexual conversion by the SCC pathway. However, IFA analysis of this parasite line using anti-Pfs16 antibodies revealed an increase in the proportion of parasites expressing this endogenous protein upon DHA exposure (*Figure 4—figure supplement 2*). The significance of this observation remains unclear but it may indicate a rapid effect of DHA on the expression of some gametocyte specific genes without further sexual development.

In these experiments, choline depletion did not increase sexual conversion via the SCC route (*Figure 4B*). This result may be explained by two alternative scenarios: (i) conversion via the SCC route is insensitive to choline depletion; (ii) ring stages are insensitive to stimulation of sexual conversion by choline depletion. To distinguish between these two possibilities, we assessed sexual conversion via the NCC pathway in cultures in which choline was depleted at different stages of the life cycle (*Figure 5A*). We found that choline depletion at the ring stage does not induce sexual conversion, in contrast to depletion at the trophozoite stage (*Figure 5B–C*). Altogether, these results show that in parasites at the ring stage neither a DHA pulse nor choline depletion induces sexual

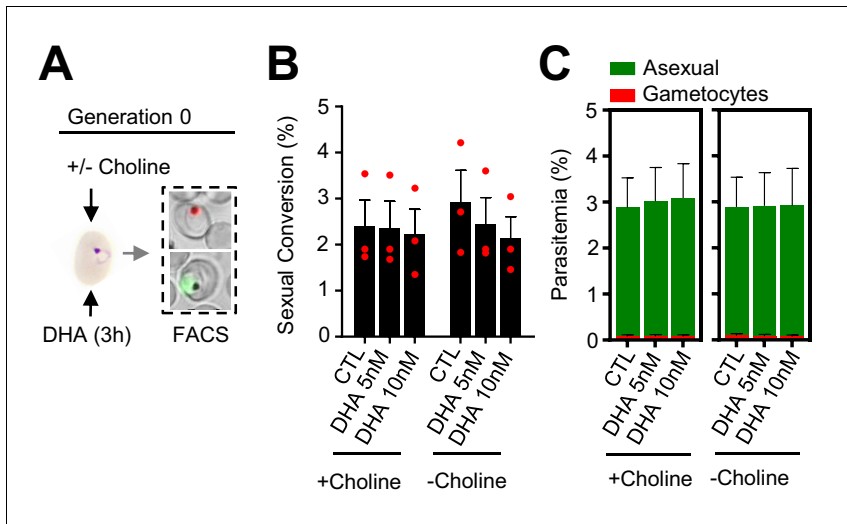

**Figure 4.** Effect of a dihydroartemisinin (DHA) pulse at the ring stage on sexual conversion by the same cycle conversion (SCC) route. (**A**) Schematic representation of the assay. Tightly synchronized cultures of the *NF54-gexp02-Tom* line under non-inducing (+ choline) or inducing (– choline) conditions were exposed to a 3 hr DHA pulse at subcurative doses at the early ring stage (0–10 hpi). Sexual conversion was measured by flow cytometry (FACS) within the same multiplication cycle (~30–40 hpi) to determine the effect of the drug pulse only on production of new gametocytes by the SSC route. (**B**) Sexual conversion rate determined by flow cytometry. No significant difference (p<0.05) with the control (no drug) was observed for any treatment condition. (**C**) Distribution of absolute parasitemia of asexual and sexual parasites (from the same flow cytometry measurements as in panel B). In all panels, data are presented as the average and s.e.m. of three independent biological replicates.

The online version of this article includes the following figure supplement(s) for figure 4:

**Figure supplement 1.** Effect of a dihydroartemisinin (DHA) pulse at the ring stage on sexual conversion by the same cycle conversion (SCC) route, determined using MitoTracker to identify viable parasites.

**Figure supplement 2.** Effect of a dihydroartemisinin (DHA) pulse at the ring stage on sexual conversion by the same cycle conversion (SCC) route in the *NF54-10.3-Tom* line, determined by three different methods.

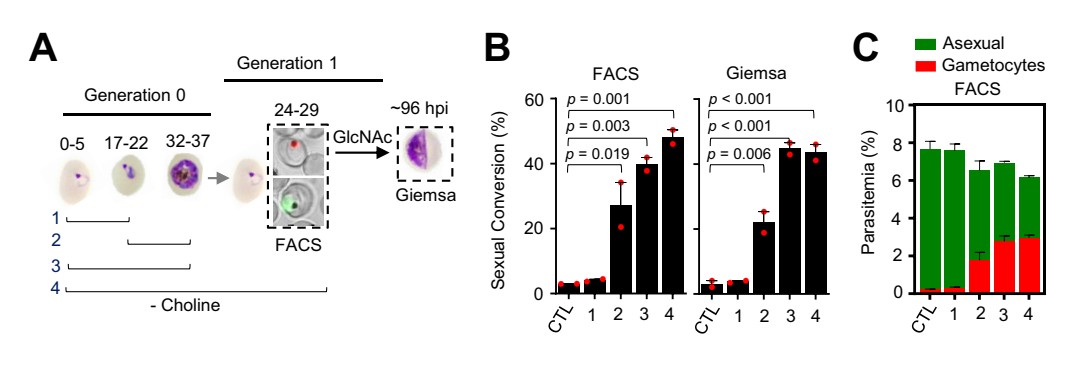

**Figure 5.** Changes in sexual conversion rates after choline depletion at different parasite stages. (**A**) Schematic representation of the assay. Choline was removed from tightly synchronized cultures of the *NF54-gexp02-Tom* line for the periods indicated, and sexual conversion rates measured after reinvasion by flow cytometry (FACS;~24–29 hpi of the following multiplication cycle) or by light microscopy analysis of Giemsa-stained smears (Giemsa;~96 hpi) in cultures treated with GlcNac. Control (CTL) cultures were maintained with choline all the time. (**B**) Sexual conversion rate for cultures under different conditions. The p-value is indicated only for choline depletion versus control significant differences (p<0.05). (**C**) Distribution of absolute parasitemia of asexual and sexual parasites, determined by flow cytometry (from the same flow cytometry measurements as in panel B). In all panels, data are presented as the average and s.e.m. of two independent biological replicates.

conversion, suggesting that this developmental stage is largely insensitive to environmental stimulation of sexual conversion.

## Exposure to CQ or heat shock at the trophozoite stage can also enhance sexual conversion

Using the same drug pulse approach, we assessed whether CQ, a drug with a different mode of action than DHA (*Haldar et al., 2018*), also stimulates sexual conversion (*Figure 6A*). Exposure to 80 nM CQ at the trophozoite stage, a dose that induces ~40% lethality (*Figure 6B*), resulted in enhanced sexual conversion rates in choline-supplemented cultures (*Figure 6C–D*; *Figure 6—figure supplement 1*). However, the level of induction was only ~ twofold, much lower than induction by DHA, and there was no consistent induction at higher or lower drug doses. Similar to DHA, CQ exposure at the trophozoite stage did not increase sexual conversion in choline-depleted cultures (*Figure 6C–D*), and exposure to CQ at the ring stage did not enhance sexual conversion by either the NCC or the SCC (*Figure 6—figure supplements 2–3*) routes. Reduced sexual conversion was observed in choline-depleted cultures treated with CQ doses that kill the vast majority of parasites, but this needs to be interpreted with caution because of the intrinsic limitations of sexual conversion assays when the majority of parasites are killed (*Figure 6*; *Figure 6—figure supplements 1–2*).

We also tested the effect of an unrelated type of stress, a 3 hr heat shock at 41.5°C mimicking a malarial febrile episode, on sexual conversion. Exposure of choline-supplemented cultures at the trophozoite stage to heat shock, which reduced survival by ~40%, resulted in a ~ fourfold increase in sexual conversion and gametocytemia (*Figure 7*).

## Enhancement of sexual conversion by DHA operates via *pfap2-g*

To determine whether stimulation of sexual conversion by DHA involves the activation of the master regulator *pfap2-g* (PF3D7_1222600), we analyzed the transcript levels for this gene after a DHA pulse, and also for one of its earliest known targets, *gexp02* (*Filarsky et al., 2018*; *Josling et al., 2020*; *Llorà-Batlle et al., 2020*; *Portugaliza et al., 2019*; *Silvestrini et al., 2010*). Transcript levels for the two genes were determined at the schizont stage of the cycle of exposure and at the ring stage of the next cycle. A subcurative DHA pulse at the trophozoite stage resulted in upregulation of both *pfap2-g* and *gexp02* relative to the *serine-tRNA ligase* (PF3D7_0717700) reference gene in choline-supplemented cultures, but not in choline-depleted cultures (*Figure 8A–C*). By contrast, exposure to DHA at the ring stage resulted in reduced expression of both genes (*Figure 8D–F*). Analysis of transcripts only 2 hr after DHA exposure at the ring stage did not reveal induction of *pfap2-g* or *gexp02* (*Figure 8G–I*), ruling out activation of the genes at a time consistent with conversion via the SCC route. Identical results were obtained when normalizing *pfap2-g* or *gexp02*

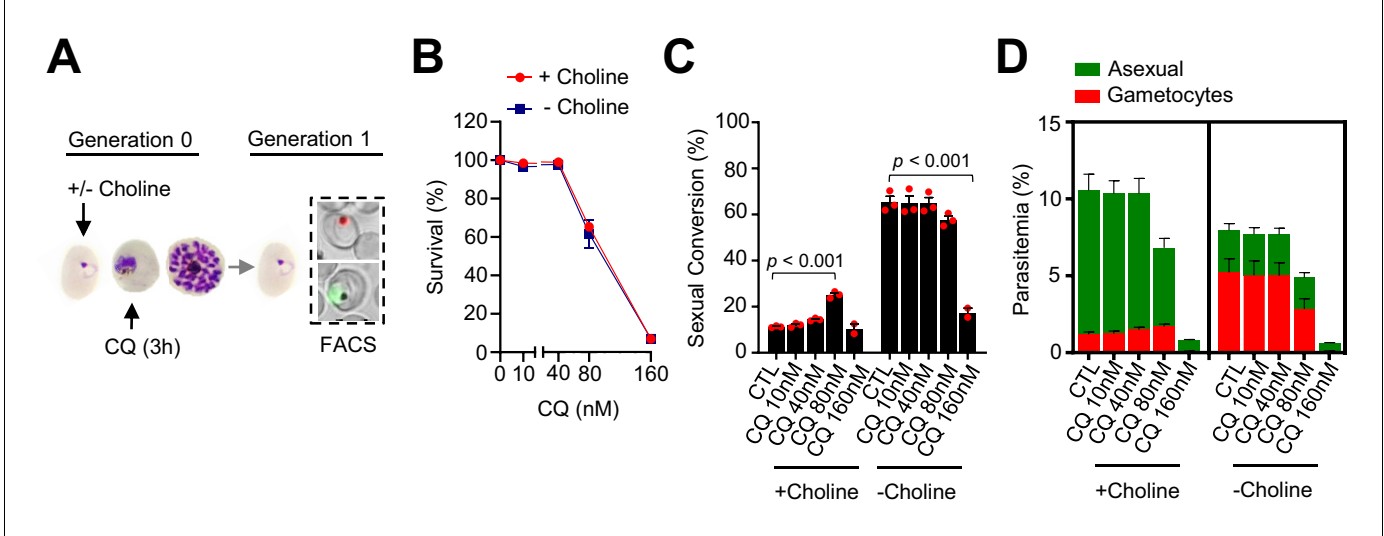

**Figure 6.** Effect of a chloroquine (CQ) pulse at the trophozoite stage on sexual conversion. (A) Schematic representation of the assay. Tightly synchronized cultures of the *NF54-gexp02-Tom* line under non-inducing (+ choline) or inducing (− choline) conditions were exposed to a 3 hr CQ pulse at subcurative doses at the trophozoite stage (25–30 hpi). Sexual conversion was measured by flow cytometry (FACS) after reinvasion (~30–35 hpi of the next multiplication cycle). (B) Survival rate of cultures exposed to the different drug doses, using total parasitemia values (asexual + sexual parasites). For each choline condition, values are presented relative to the parasitemia in the control cultures (no drug). (C) Sexual conversion rate determined by flow cytometry. The p-value is indicated only for treatment versus control (no drug) significant differences (p<0.05). (D) Distribution of absolute parasitemia of asexual and sexual parasites (from the same flow cytometry measurements as in panel C). In all panels, data are presented as the average and s.e.m. of three independent biological replicates.

The online version of this article includes the following figure supplement(s) for figure 6:

**Figure supplement 1.** Effect of a chloroquine (CQ) pulse at the trophozoite stage on sexual conversion, determined using MitoTracker to identify viable parasites.

**Figure supplement 2.** Effect of a chloroquine (CQ) pulse at the ring stage on sexual conversion.

**Figure supplement 3.** Effect of a chloroquine (CQ) pulse at the ring stage on sexual conversion by the same cycle conversion (SCC) route.

transcript levels against *ubiquitin-conjugating enzyme* (PF3D7_0812600) as a reference gene (*Figure 8—figure supplement 1*). Overall, the findings of these transcriptional analyses clearly mirror the effect of the drug on sexual conversion rates, indicating that induction of sexual conversion by DHA is associated with *pfap2-g* activation.

## Discussion

ARTs are the key component of ACTs, the most widely used treatment for clinical malaria. Additionally, ACTs may be widely administered in mass drug administration campaigns aimed at malaria elimination. Given that the success of malaria control and elimination efforts largely depends on preventing disease transmission, understanding the impact of ARTs on the production of transmission forms is of paramount importance. Our results show a complex effect of DHA on the trade-off between asexual proliferation and the formation of transmission forms. Exposure of parasites at the trophozoite stage to subcurative doses of DHA resulted in a large increase in sexual conversion rates and total number of gametocytes, which were viable and infectious to mosquitoes. However, this was not observed when parasites were exposed to the same drug doses at the ring stage. Furthermore, in cultures in which sexual conversion was already stimulated at the metabolic level (i.e., by depletion of choline), DHA did not further stimulate sexual conversion at either stage. The accurate determination of the impact of DHA on sexual conversion rates at different stages was possible thanks to the development of an assay that uses a short drug pulse and reporter parasite lines that enable very early detection of gametocytes by flow cytometry. A limitation of our study is that we only used wild type parasite lines that do not carry mutations in the PfKelch13 protein associated with artemisinin resistance (*Ariey et al., 2014*; *Birnbaum et al., 2020*; *Yang et al., 2019*). Future

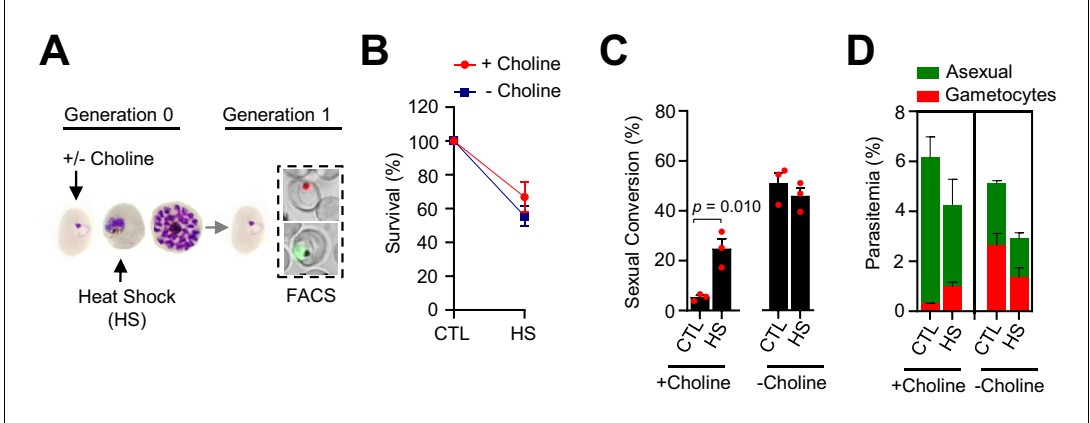

**Figure 7.** Effect of heat shock at the trophozoite stage on sexual conversion. (**A**) Schematic representation of the assay. Tightly synchronized cultures of the *NF54-gexp02-Tom* line under non-inducing (+ choline) or inducing (– choline) conditions were exposed to a 3 hr heat shock (41.5˚C) at the trophozoite stage (25–30 hpi). Sexual conversion was measured by flow cytometry (FACS) after reinvasion (~30–35 hpi of the next multiplication cycle). (**B**) Survival rate of cultures exposed to heat shock (HS) or maintained at 37˚C (CTL), using total parasitemia values (asexual + sexual parasites). For each choline condition, values are presented relative to the parasitemia in the control cultures. (**C**) Sexual conversion rate determined by flow cytometry. The p-value is indicated only for heat shock versus control significant differences (p<0.05). (**D**) Distribution of absolute parasitemia of asexual and sexual parasites (from the same flow cytometry measurements as in panel C). In all panels, data are presented as the average and s.e.m. of three independent biological replicates.

studies should assess the effect of DHA on sexual conversion in parasite lines carrying such mutations, to determine whether or not the effect of the drug on sexual conversion is linked to its effect on parasite survival.

The overall impact of a drug on the transmission potential of an infection depends on its effect on the sexual conversion rate, and on several other factors. In the case of ARTs, the stimulation of sexual commitment at the trophozoite stage may not result in an overall increase in transmission due to rapid clearance of asexual parasites, which prevents new rounds of gametocyte production, and to the activity of the drug against developing and mature gametocytes. Indeed, several studies have observed that treatment with drug combinations containing ARTs reduce gametocyte density and the duration of gametocyte carriage (*Bousema et al., 2006*; *Bousema et al., 2010*; *Ippolito et al., 2017*; *Karl et al., 2015*; *Okell et al., 2008*; *Price et al., 1996*; *Sawa et al., 2013*; *Targett et al., 2001*; *von Seidlein et al., 2001*; *WWARN Gametocyte Study Group, 2016*). Notwithstanding the net reduction of transmission potential commonly observed after ART treatment, it is possible that patients in which many of the parasites are at the trophozoite stage at the time of ART administration may experience a peak of circulating gametocytes ~ 10 days after treatment (the time required for gametocyte maturation), if the drug does not kill all parasites. In this regard, it is noteworthy that the largest stimulation of sexual conversion was observed at subcurative doses of the drug. Such low drug concentrations may occur during treatment with substandard or underdosed drugs, through poor compliance with the prescribed regimen, as a consequence of drug malabsorption, or as the drug is eliminated following its natural pharmacokinetics profile. Treatment associated with low ARTs concentration may enable survival of some parasites, and at the same time enhance the probability of sexual conversion. Thus, our findings have potential public health implications for the use of ARTs in treatment and elimination strategies. While the benefits of ARTs for malaria treatment clearly outweigh the potential risks, the possibility that ARTs increase the transmission potential of some patients should be taken into account when considering their massive use in preventive treatment or elimination campaigns.

There is ongoing debate regarding whether human malaria parasites can modulate their level of investment in producing transmission forms as a response to 'stress' (i.e., a condition that reduces the asexual multiplication rate). Whether the impact of stress on sexual conversion rates is positive (enhancement) or negative (reduction) also remains controversial (*Buckling et al., 1999a*; *Buckling et al., 1999b*; *Buckling et al., 1997*; *Koepfli and Yan, 2018*; *Peatey et al., 2009*; *Schneider et al., 2018*). Evolutionary theory for life histories predicts that treatment with low doses

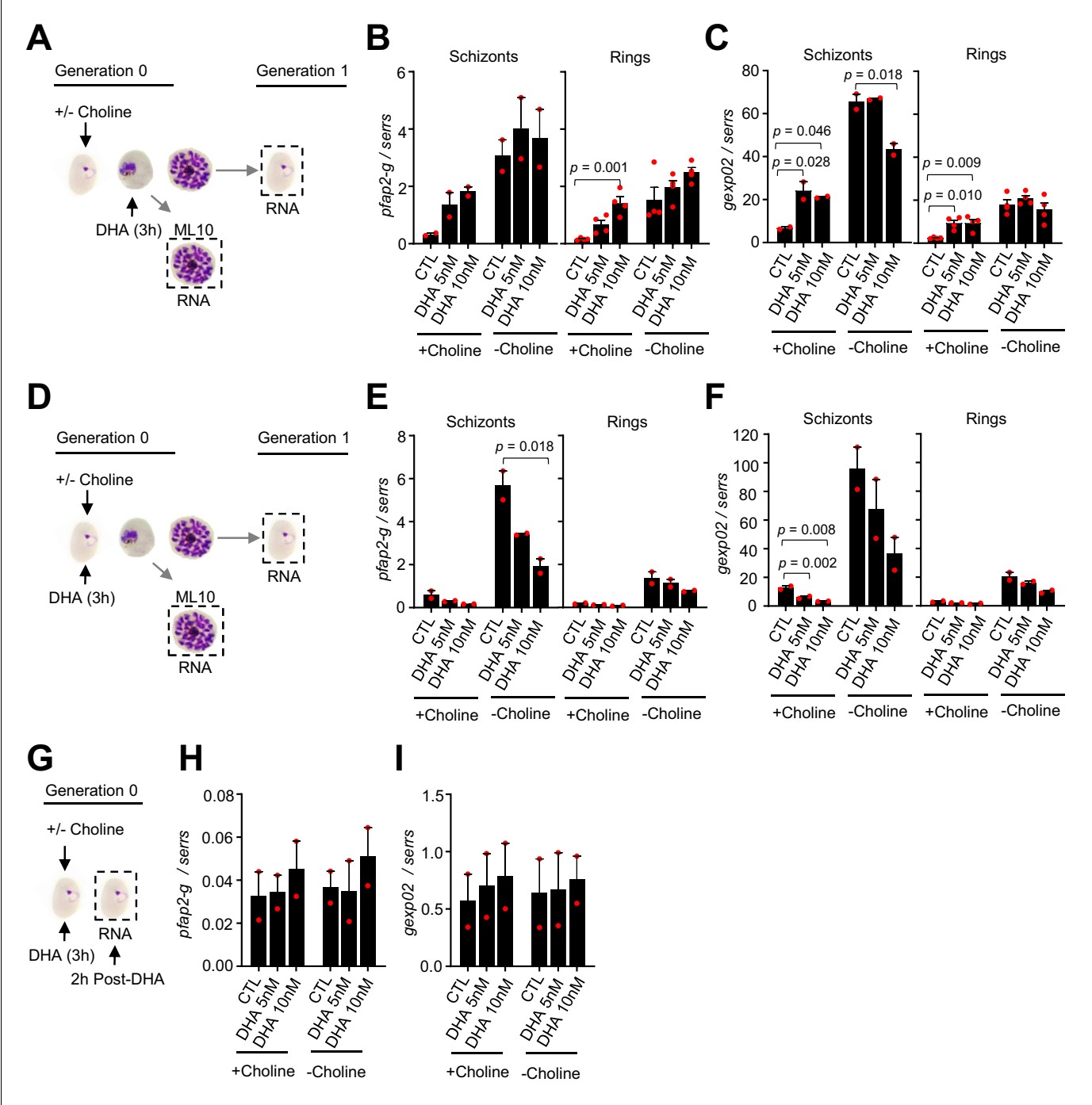

**Figure 8.** Changes in the expression of *pfap2-g* and *gexp02* after a dihydroartemisinin (DHA) pulse. (**A**) Schematic representation of the assay. Tightly synchronized cultures of the *NF54-gexp02-Tom* line under non-inducing (+ choline) or inducing (– choline) conditions were exposed to a 3 hr DHA pulse at subcurative doses at the trophozoite stage (25–30 hpi). RNA for transcriptional analysis was collected from ML10-treated cultures at the mature schizont stage (48–53 hpi) and, after reinvasion, from cultures at the early ring stage (cultures not treated with ML10,~5 hpi). (**B–C**) Transcript levels of *pfap2-g* (**B**) or *gexp02* (**C**) normalised against the *serine-tRNA ligase* (*serrs*) gene. (**D–F**) Same as panels A-C, but cultures were exposed to DHA at the ring stage (0–10 hpi). (**G–I**) Same as panels D-F, but RNA for transcriptional analysis was collected only 2 hr after completing the drug pulse. Data are presented as the average and s.e.m. of four (panels B-C, rings) or two (other panels) independent biological replicates. The p-value is indicated only for treatment versus control (no drug) significant differences (p<0.05).

The online version of this article includes the following figure supplement(s) for figure 8:

**Figure supplement 1.** Changes in the transcript levels of *pfap2-g* and *gexp02* after a dihydroartemisinin (DHA) pulse, normalized against the *uce* gene.

of antimalarial drugs results in reproductive restraint (reduced sexual conversion) to facilitate within-host survival, whereas treatment with high doses that kill the majority of the parasites elicits terminal investment (increased sexual conversion). The results of a recent study using a murine model of malaria were consistent with this prediction (*Schneider et al., 2018*). Our experiments with in vitro cultured *P. falciparum* exposed to low doses of DHA at the ring stage were also consistent with these prediction, as this resulted in moderately reduced sexual conversion rates. In contrast, experiments in which a pulse of DHA or CQ at a low dose was administered at the trophozoite stage showed the opposite trend, such that the subcurative treatment stimulated sexual conversion. This latter result is in line with some previous studies using *P. falciparum* (*Buckling et al., 1999b*; *Peatey et al., 2009*) or a murine malaria model (*Buckling et al., 1999a*; *Buckling et al., 1997*). A possible explanation for the discrepancy with the predictions of evolutionary theory (when exposure occurs at the trophozoite stage) is that in the absence of stress, sexual conversion in *P. falciparum* is already restrained, with estimated conversion rates of ~1% in human infections (*Cao et al., 2019*; *Eichner et al., 2001*). Thus, a further reduction of the investment in transmission upon exposure to low drug doses would not have a substantial impact on within-host survival, implying that this response would not provide a selective advantage, whereas the opposite response can enhance the chances of transmission. Of note, the absence of LysoPC and choline (*Brancucci et al., 2017*), or heat shock, all of which reduce the multiplication rate of *P. falciparum* cultures and therefore can also be considered as sublethal stress signals, also stimulate sexual conversion. Together, the results of experiments with *P. falciparum* cultures exposed to low level of stress at the trophozoite stage do not support the predictions of evolutionary theory, whereas for murine malaria parasites different studies reported conflicting results. In this regard, it is possible that different *Plasmodium* species use different strategies to adjust sexual conversion rates upon stress: although the role of AP2-G as the master regulator of sexual conversion appears to be widely conserved in all malaria parasite species, upstream events involved in the regulation of sexual conversion are remarkably different between human and murine parasites. The latter show higher conversion rates, do not alter sexual conversion in response to LysoPC restriction, and their genomes lack a *gdv1* ortholog (*Ngotho et al., 2019*).

Our experiments establish that sexual conversion can be stimulated by exposure to DHA at the trophozoite stage, but not at the early ring stage. Of note, stimulation of sexual conversion by depletion of choline (as a proxy for LysoPC depletion) or by exposure to CQ shows a similar stage dependency, suggesting that the ring stage is largely insensitive to stimulation of sexual conversion. At the ring stage, some types of stress, such as exposure to DHA, may induce latency of a rather small fraction of the parasites as a means of population survival (*Barrett et al., 2019*; *Talman et al., 2019*), rather than enhancing sexual conversion. Furthermore, we found that in cultures in which sexual conversion is stimulated by choline depletion, it cannot be further stimulated by drugs, such that there are no additive or synergistic effects between drugs and choline depletion. Together, these observations suggest that the different stimuli converge into the same mechanism of *pfap2-g* activation, which likely involves cellular components that are absent during the ring stage. Because stimulation of sexual conversion by choline depletion has been shown to involve GDV1 (*Filarsky et al., 2018*), which is only expressed in the second half of the intraerythrocytic development cycle and is absent from ring stage parasites, we hypothesize that stimulation by DHA may also depend on GDV1. A possible explanation for the similar effects of DHA and choline depletion on sexual conversion is that treatment with DHA may result in choline depletion: DHA induces damage on membrane lipids (*Hartwig et al., 2009*), which may increase the use of LysoPC or choline, resulting in a reduction of their levels. However, heat shock, a completely different type of stress that is not known or predicted to affect LysoPC or choline levels, also enhanced sexual conversion. This result suggests that stimulation of sexual conversion can occur without the involvement of choline metabolism. As an alternative model, parasites may be able to sense a state of mild to moderate 'stress' or growth restriction (*Schneider et al., 2018*): the drug doses that result in increased sexual conversion, as well as LysoPC or choline restriction (*Brancucci et al., 2017*; *Portugaliza et al., 2019*) and heat shock, were all associated with a < 50% reduction of growth rates. The observation that a 2 nM pulse of DHA, which does not have any measurable effect on parasite survival, appears to stimulate sexual conversion may indicate that sexual conversion is triggered by the response associated with moderate stress, rather than by growth restriction per se. In this regard, activation of the cellular stress response has been proposed to be associated with enhanced gametocyte production

(*Chaubey et al., 2014*), and DHA triggers such a stress response (*Bridgford et al., 2018*; *Zhang et al., 2017*). Further research is needed to establish the molecular mechanisms underlying the modulation of sexual conversion rates by different environmental conditions.

Altogether, here we provide a detailed characterization of the changes in *P. falciparum* sexual conversion rates that occur in response to a pulse of DHA. We demonstrate remarkable plasticity in sexual conversion rates, and a complex response that depends on the stage of the parasites at the time when they are exposed to the drug, the drug dose, and the metabolic state (presence or absence of choline). This complex scenario may explain the discrepant results obtained by previous studies. The assay that we have developed to test the impact of DHA on sexual conversion rates can be used to test the impact of any other drug or condition, as shown here for CQ and heat shock. Of note, the success of malaria elimination efforts largely depends on the ability to reduce or interrupt transmission. Although our results are not of immediate public health concern because the overall impact of treatment with ACTs is a reduction of the transmission potential, at least when compared with other drugs, the capacity of ARTs to induce sexual conversion must be taken into account. Otherwise, under certain conditions, treatment may result in an increase in transmission that could jeopardize efforts to eliminate malaria.

# Materials and methods

## Key resources table

| Reagent type (species) or resource | Designation | Source or reference | Identifiers | Additional information |
|---|---|---|---|---|
| Gene (*Plasmodium falciparum*) | *pfap2-g* | PlasmoDB | PF3D7_1222600 | |
| Gene (*Plasmodium falciparum*) | *gexp02* | PlasmoDB | PF3D7_1102500 | |
| Cell line (*Plasmodium falciparum*) | NF54-gexp02-Tom | PMID:31601834 | | Maintained in culture with 2 mM choline |
| Cell line (*Plasmodium falciparum*) | E5-gexp02-Tom | PMID:31601834 | | Maintained in culture with 2 mM choline |
| Cell line (*Plasmodium falciparum*) | NF54-10.3-Tom | PMID:31601834 | | Maintained in culture with 2 mM choline |
| Commercial assay or kit | RNeasy Mini Kit | Qiagen | Cat. No. 74104 | |
| Chemical compound, drug | ML10 | PMID:28874661; S. Osborne (LifeArc) and D. Baker (LSHTM) | | cGMP-dependent protein kinase inhibitor |
| Chemical compound, drug | Dihydroartemisinin (DHA) | Sigma-Aldrich | Cat. No. D7439 | |
| Chemical compound, drug | Chloroquine | Sigma-Aldrich | Cat. No. C6628 | |
| Chemical compound, drug | Choline chloride | Sigma-Aldrich | Cat. No. C7527 | |
| Chemical compound, drug | N-acetyl-d-glucosamine | Sigma-Aldrich | Cat. No. A8625 | |

*Continued on next page*

*Continued*

| Reagent type (species) or resource | Designation | Source or reference | Identifiers | Additional information |
|---|---|---|---|---|
| Software, algorithm | BD FACSDiva Software | BD Biosciences | RRID:SCR_001456 | Flow cytometry acquisition and analysis using BD LSRFortessa machine |
| Software, algorithm | Flowing Software version 2.5.1 | Perttu Terho | RRID:SCR_015781 | Flow cytometry data analysis |
| Software, algorithm | Prism 8 | GraphPad | RRID:SCR_002798 | |
| Antibody | Pfs16 (mouse, monoclonal) | R.Sauerwein, Radboud University | 32F717:B02 | IFA (1:400) |
| Antibody | Goat-anti-mouse IgG–Alexa Fluor 488 | Thermo Fisher | Cat. No. A11029 | IFA (1:1000) |
| Other | SYTO 11 | Life Technologies | Cat. No. S7573 | Flow cytometry (0.016 µM) |
| Other | MitoTracker Deep Red FM | Invitrogen | Cat. No. M22426 | Flow cytometry (0.6 µM) |
| Other | DAPI | Applichem lifescience | Cat. No. A4099.0005 | IFA (5 µg/mL) |
| Other | Hoechst 33258 | Thermo Fisher | Cat. No. H3569 | Live cell fluorescence microscopy (2 µM) |

## Parasite cultures

The transgenic reporter lines *NF54-gexp02-Tom*, *E5-gexp02-Tom*, and *NF54-10.3-Tom* were previously described and characterized (*Portugaliza et al., 2019*). These parasite lines carry a *tdTomato* reporter gene under the control of either the *gexp02* or the *etramp10.3* promoters. Since these *P. falciparum* lines were generated and validated in our laboratory, and the expression pattern of the fluorescent markers confirms their identities, additional authentication was considered unnecessary. They were not tested for *Mycoplasma*, but *Mycoplasma* contamination is not known to affect any of the parameters analyzed in this study.

Cultures were regularly maintained at 37°C under shaking (100 rpm) or static conditions in a hypoxic atmosphere (2% $O_2$, 5.5% $CO_2$, balance $N_2$), with B+ erythrocytes (3% hematocrit) and standard RPMI-HEPES parasite culture medium containing 0.5% Albumax and supplemented with 2 mM choline (*Filarsky et al., 2018*; *Portugaliza et al., 2019*). Erythrocytes were obtained from the Catalan official blood bank (Banc de Sang i Teixits). To obtain cultures of a well-defined age window, we used Percoll/sorbitol synchronization. In brief, Percoll-purified schizonts were used to establish a fresh culture that 5 or 10 hr later was subjected to 5% D-sorbitol lysis to obtain cultures of a defined 0–5 or 0–10 hpi age window. For heat shock experiments, synchronized cultures at the trophozoite stage were transferred for 3 hr to an incubator at 41.5°C and then placed back at 37°C (*Rovira-Graells et al., 2012*).

Cultures for the production of mature gametocytes for mosquito infection were maintained in a semi-automated shaker incubator system as described (*Mogollon et al., 2016*). Fresh human serum and erythrocytes for these experiments were obtained from the Dutch National Blood Bank (Sanquin Amsterdam, the Netherlands; permission granted from donors for the use of blood products for malaria research and microbiology; tested for safety). Erythrocytes and human serum from different donors were pooled.

## Drug treatment and determination of drug survival rates

To test the impact of drugs on sexual conversion in the presence or absence of choline, after tight synchronization (0–5 or 0–10 hpi) cultures at ~1.5% parasitemia were split in two and one culture was maintained with a 2 mM choline supplement whereas the other had no choline added. Drug pulses with DHA (Sigma-Aldrich no. D7439) or CQ (Sigma-Aldrich no. C6628) were performed at 1–6 hpi (*NF54-10.3-Tom*) or 0–10 hpi (*NF54-gexp02-Tom*) for exposure at the ring stage, or at 25–30 hpi for exposure at the trophozoite stage. After 3 hr, the drug was removed and fresh, pre-warmed culture medium was added. In some experiments, 200 nM DHA was maintained for 48 hr as a 'kill' control (*Xie et al., 2014*).

The survival rate was calculated as the total parasitemia (asexual + sexual parasites) at the next cycle after drug exposure (measured at ~30–35 hpi or ~30–40 hpi) in treated cultures relative to control cultures, and expressed as percentage. Parasitemia was measured by flow cytometry (see below).

## Determination of sexual conversion rates

A schematic of our approach to determine sexual conversion rates is provided in *Figure 1—figure supplement 1*. We define day 0 (D0) as the first day of the next cycle after drug exposure, which corresponds to the first day of Generation one in the schematics in the figures. D1 corresponds to the day when new sexual parasites become stage I gametocytes. When using the *NF54-gexp02-Tom* and *E5-gexp02-Tom* lines, the sexual conversion rate was calculated as the sexual stage parasitemia divided by the total (sexual + asexual) parasitemia, and expressed as percentage. Asexual and sexual parasites were quantified by flow cytometry at ~30–35 hpi or ~30–40 hpi (D1) of the cycle after drug treatment, in the absence of chemicals that inhibit asexual replication. When using the *NF54-10.3-Tom* line, the sexual conversion rate was measured by dividing the gametocytemia on day 3 (D3) by the initial ring stage parasitemia on D0. In this case, cultures were treated with 50 mM N-acetyl-D-glucosamine (GlcNAc; Sigma-Aldrich no. A8625) from D1 onwards to inhibit asexual replication. In experiments with the *NF54-10.3-Tom* line, gametocytemia was also measured on D0 to identify gametocytes already present in the culture at the beginning of the experiment, but it was found to be negligible. In any case, it was subtracted from D3 gametocytemia, such that only gametocytes newly formed during the assay were considered. Unless otherwise indicated, statistical analysis of differences in sexual conversion was performed using one-way ANOVA with Tukey HSD as the post hoc test. Variance was assumed to be homogenous because the sample size was the same for all groups, and they contain the same type of data.

## Flow cytometry

Flow cytometry analysis to measure parasitemia at the cycle of drug exposure was performed using the nucleic acid stain SYTO 11 (0.016 µM) (Life Technologies no. S7573) and a BD FACSCalibur machine as previously described (*Rovira-Graells et al., 2016*). To measure asexual and tdTomato-positive sexual parasites, we used a BD LSRFortessa machine as previously described (*Portugaliza et al., 2019*), with small modifications after the addition of the mitochondrial membrane potential MitoTracker Deep Red FM fluorescent dye (Invitrogen no.M22426) at 0.6 µM to identify live parasites (*Figure 1—figure supplements 7–8*; *Amaratunga et al., 2014*). Briefly, the erythrocyte population was defined using the side scatter area (SCC-A) versus forward scatter area (FSC-A) plot, followed by singlet gating using the forward scatter height (FSC-H) versus FSC-A plot. From the singlet population, the parasites were simultaneously analyzed for tdTomato fluorescence (laser: 561 nm; filter: 582/15; power: 50 mW), SYTO 11 fluorescence (laser: 488 nm; Filter: 525/50-505LP; power: 50 mW), and MitoTracker fluorescence (laser: 640 nm; Filter: 670/14-A; power: 40 mW). Total gametocytes were quantified on the double-positive gate of the tdTomato versus SYTO 11 plot. Total asexual stages were quantified on the tdTomato-negative but SYTO 11-positive gate, whereas viable asexual stages were measured on the tdTomato-negative but MitoTracker-positive gate. Flowing Software version 2.5.1 (Perttu Terho) was used for downstream analysis.

## Immunofluorescence assay

Immunofluorescence assays (IFA) were performed as previously described (*Bancells et al., 2019*; *Portugaliza et al., 2019*). Briefly, an aliquot of culture was treated with 80 nM ML10 (cGMP-

dependent protein kinase inhibitor) (*Baker et al., 2017*), starting at ~30–35 hpi until ~48–53 hpi, to inhibit schizont rupture and allow maturation of gametocytes to the stage when all of them express Pfs16. Air-dried blood smears containing schizonts and stage I gametocytes (~48–53 hpi) were fixed with 1% paraformaldehyde in PBS, permeabilized with 0.1% Triton X-100 in PBS, and blocked with 3% BSA in PBS. The gametocyte-specific primary antibody mouse-anti-PfS16 (1:400; 32F717:B02, a gift from R. Sauerwein, Radboud University) and the goat-anti-mouse IgG–Alexa Fluor 488 secondary antibody (1:1,000, Thermo Fisher no.A11029) were used to identify stage I gametocytes, whereas DAPI (5 µg/mL) was added to stain parasite DNA. IFA slides were mounted using Vectashield (Palex Medical) and viewed under an Olympus IX51 epifluorescence microscope for determination of sexual conversion rates. A minimum of 200 DAPI-positive cells was counted for each sample.

## Transcriptional analysis

Trizol reagent (Invitrogen no. 15596026) was used to collect and preserve total RNA, followed by extraction using a protocol designed for samples with low RNA concentration (*Mira-Martínez et al., 2017*). Briefly, RNA from Trizol samples was purified using a commercial kit (RNeasy Mini Kit, Qiagen no. 74104) with additional on-column DNAse treatment (Qiagen no. 79254). Next, cDNA synthesis was performed using the AMV Reverse Transcription System (Promega), with a combination of oligo (dT) and random primers. Quantitative PCR (qPCR) analysis of the cDNAs was performed as previously described (*Bancells et al., 2019*) using triplicate wells (technical replicates) for each biological replicate of each sample. Transcript levels of *pfap2-g* and *gexp02* were normalized against the housekeeping genes *serine-tRNA ligase* and *ubiquitin-conjugating enzyme*. All qPCR primers used have been previously described (*Bancells et al., 2019*; *Portugaliza et al., 2019*). Statistical analysis of transcript levels was performed using one-way ANOVA with Tukey HSD as the post hoc test, as for the analysis of sexual conversion rates.

## Production of mature gametocytes and mosquito feeding

Cultures maintained in a medium containing 0.5% Albumax and supplemented with 2 mM choline were synchronized for ring stages by D-Sorbitol treatment and diluted to a final parasitemia of 1.5%. At 22 hr after synchronization, DHA (5 nM) was added to the cultures for 3 hr, and 24 hr later (i.e., after reinvasion) culture conditions were changed to medium with 10% human serum instead of Albumax and choline, and GlcNac (50 mM) was added to kill asexual stages. GlcNac was maintained for 4 d. Gametocyte cultures were followed during 9–13 d after DHA-treatment with medium changes twice a day, but without replenishing with fresh erythrocytes. At days 9–13, gametocyte development was analyzed in Giemsa stained blood smears and exflagellation was monitored after activation as described (*Marin-Mogollon et al., 2018*). Gametocytes (day 10–13) were fed to *Anopheles stephensi* mosquitoes using the standard membrane feeding assay (SMFA) (*Marin-Mogollon et al., 2018*; *Ponnudurai et al., 1989*). Oocysts (day 7 and 14) and salivary gland sporozoites (day 14) were counted as described (*Marin-Mogollon et al., 2018*). Statistical analysis of differences in the parameters measured (*Figure 2*) was performed using one-way ANOVA with Tukey HSD as the post hoc test, except for oocyst/mosquito values. For the oocyst/mosquito analysis, we used the Kruskal-Wallis test with post hoc Dunn's test (this is used because the data is not normally distributed, Shapiro-Wilk test $p < 0.001$).

## Acknowledgements

We are grateful to Robert W Sauerwein (Radboud University, The Netherlands) for the anti-Pfs16 monoclonal antibody, and to Simon Osborne (LifeArc, UK) and David Baker (LSHTM, UK) for providing the compound ML10 and advice on its use. We thank Oriol Llorà-Batlle (ISGlobal) for help setting gametocyte experiments, Elisabet Tintó-Font (ISGlobal) for help setting the heat shock experiments, Blandine M Franke-Fayard and Severine Chevalley (Leiden University Medical Center) for support with gametocyte cultures and mosquito infections, and the Flow Cytometry core facility of the IDI-BAPS for technical help. This work was supported by a grant from the Spanish Ministry of Economy and Competitiveness (MINECO)/Agencia Estatal de Investigación (AEI) [SAF2016-76190-R to AC], co-funded by the European Regional Development Fund (ERDF, European Union). ITM, UvA, and ISGlobal are members of the TransGlobalHealth–Erasmus Mundus Joint Doctorate Programme, European Union (scholarship number 2016–1346 to HPP). This research is part of ISGlobal's Program

on the Molecular Mechanisms of Malaria, which is partially supported by the Fundación Ramón Areces. We acknowledge support from the Spanish Ministry of Science and Innovation through the 'Centro de Excelencia Severo Ochoa 2019–2023' Program (CEX2018-000806-S), and support from the Generalitat de Catalunya through the CERCA Program.

## Additional information

### Funding

| Funder | Grant reference number | Author |
|---|---|---|
| Ministerio de Economía y Competitividad | SAF2016-76190-R | Alfred Cortes |
| European Commission | 2016-1346 | Harvie P Portugaliza<br>Christopher Pell<br>Anna Rosanas-Urgell<br>Alfred Cortes |

The funders had no role in study design, data collection and interpretation, or the decision to submit the work for publication.

### Author contributions

Harvie P Portugaliza, Conceptualization, Resources, Formal analysis, Investigation, Methodology, Writing - original draft, Writing - review and editing; Shinya Miyazaki, Fiona JA Geurten, Investigation, Writing - review and editing; Christopher Pell, Chris J Janse, Supervision, Writing - review and editing; Anna Rosanas-Urgell, Conceptualization, Supervision, Writing - review and editing; Alfred Cortés, Conceptualization, Resources, Supervision, Funding acquisition, Writing - original draft, Project administration, Writing - review and editing

### Author ORCIDs

Harvie P Portugaliza https://orcid.org/0000-0003-3038-6699
Alfred Cortés https://orcid.org/0000-0003-0730-6582

### Decision letter and Author response

Decision letter https://doi.org/10.7554/eLife.60058.sa1
Author response https://doi.org/10.7554/eLife.60058.sa2

## Additional files

### Supplementary files

• Transparent reporting form

### Data availability

All data generated or analysed during this study are included in the manuscript and supporting files.

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
