## [Decision Letter]

**Acceptance summary:**

Transmission of malaria parasites from humans to mosquitoes depends on the formation of sexual blood stages, the gametocytes. Here, Portugaliza et al., capitalize on reporter parasite lines expressing fluorescence markers in early gametocytes to measure the effects of artemisinin, the front-line antimalarial drug, on *Plasmodium falciparum* sexual conversion. The study reveals that exposure to subcurative doses of dihydroartemisinin stimulates parasite sexual conversion, in a dose and stage-dependent manner. This work illustrates the complex interplay between drug exposure, parasite stage and metabolic conditions in the trade-off between asexual proliferation and formation of transmission forms.

**Decision letter after peer review:**

Thank you for submitting your article "Artemisinin exposure at the trophozoite stage enhances sexual conversion in the malaria parasite *Plasmodium falciparum*" for consideration by *eLife*. Your article has been reviewed by three peer reviewers, one of whom is a member of our Board of Reviewing Editors, and the evaluation has been overseen by Dominique Soldati-Favre as the Senior Editor. The following individuals involved in review of your submission have agreed to reveal their identity: Catherine Lavazec (Reviewer #3).

The reviewers have discussed the reviews with one another and the Reviewing Editor has drafted this decision to help you prepare a revised submission.

Summary:

This is an interesting and well-presented study showing that exposure to subcurative doses of antimalarial drugs stimulates sexual conversion in the malaria parasite *Plasmodium falciparum*.

The authors exploited recently published gametocyte-reporter parasite lines to measure the impact of drug treatment on sexual conversion in *P. falciparum* cultures. They found that exposure to subcurative doses of dihydroartemisinin at the trophozoite stage results in an increase in sexual conversion, mimicking depletion of choline in the culture medium. This effect is associated with an upregulation of AP2G, the master regulator of sexual conversion. Sexual conversion is also enhanced by exposure to another antimalarial drug, chloroquine, albeit to a lesser extent. In contrast, the ring stage appears to be insensitive to sexual conversion stimulation by artemisinin or choline depletion.

Stress, including drug-induced stress, is known to stimulate sexual conversion in *P. falciparum*, so the conceptual novelty is somewhat limited. While the meaning of such a phenomenon in a clinical context remains unclear, the data are solid and this study provides a novel tool to investigate the signaling pathways leading to sexual conversion.

Essential revisions:

1) Effect on ring stages

A main novel finding here is that ring stages of Pf are refractory to sexual conversion following drug and metabolic changes. DHA arrests ring stages before killing them, so DHA-treated rings might require a longer latent period before conversion to gametocyte. The authors should measure conversion at least 24-48hours after their measurement for DHA-treated trophozoites. In Figure 2, Did the authors test other concentrations of DHA on rings? Since they show that the DHA effect at the trophozoite stage is dose-dependent, one may expect that DHA affects the sexual conversion at the ring stage with concentrations higher than 10 nM.

2) Assessment of sexual conversion.

It is not clear how and when the total % of gametocytes was measured following drug treatments (data presented in Figure 1E and Figure 2E) and whether DHA increases both the proportion of early gametocyte and subsequent development into mature gametocyte after 10-14 days or if this increase was driven merely by a reduction in asexual forms or increased RBC lysis following drug exposure. In this study, and in the previous article by Portugaliza et al., (2019), the authors define the sexual conversion rate as the proportion of parasites that convert into sexual forms at each replicative cycle, which can be estimated as the gametocytemia divided by the initial parasitemia. This is indeed how the authors calculated the sexual conversion rate when using the NF54-10.3-Tom line (measured by dividing the gametocytemia on day 3 (D3) by the initial ring stage parasitemia on D0). However, when using the NF54-gexp02-Tom reporter line, the authors used another calculation (subsection “Determination of sexual conversion rates”) ("the sexual conversion rate was calculated as the sexual stage parasitemia divided by the total (sexual + asexual) parasitemia, and expressed as percentage"). This corresponds to the proportion of gametocytes and not exactly to the sexual conversion rate, which is over-estimated when the number of asexual parasites is reduced due to the effects of the drug. Therefore, the axis label in graphs 1D, 2D, 3B, 4B, 5C, etc…should be changed to something like "proportion of gametocytes". The authors should consider using a more accurate calculation of the sexual conversion, as they did in their 2019 paper with the NF54-gexp02-Tom parasites or in this study with the NF54-10.3-Tom.

3) A major question is whether DHA exposure induction of sexual conversion is via a choline-dependent or independent pathway. Choline depletion induces sexual conversion and may present as a confounding effect in DHA treated trophozoites since DHA alkylates lipids, especially in trophozoites. Choline measurements should have been done in the DHA or CQ treated parasites.

4) It seems that only one single transmission experiment was performed (Supplementary figure 5), with a very small number of mosquitoes. The authors should repeat this experiment to strengthen the conclusion that ART enhances production of functional gametocytes.

---

## [Author Response]

Summary:This is an interesting and well-presented study showing that exposure to subcurative doses of antimalarial drugs stimulates sexual conversion in the malaria parasite Plasmodium falciparum.The authors exploited recently published gametocyte-reporter parasite lines to measure the impact of drug treatment on sexual conversion in P. falciparum cultures. They found that exposure to subcurative doses of dihydroartemisinin at the trophozoite stage results in an increase in sexual conversion, mimicking depletion of choline in the culture medium. This effect is associated with an upregulation of AP2G, the master regulator of sexual conversion. Sexual conversion is also enhanced by exposure to another antimalarial drug, chloroquine, albeit to a lesser extent. In contrast, the ring stage appears to be insensitive to sexual conversion stimulation by artemisinin or choline depletion.Stress, including drug-induced stress, is known to stimulate sexual conversion in P. falciparum, so the conceptual novelty is somewhat limited. While the meaning of such a phenomenon in a clinical context remains unclear, the data are solid and this study provides a novel tool to investigate the signaling pathways leading to sexual conversion.

We thank the reviewers for the careful review, for the important suggestions to improve the manuscript and for appreciating the interest and quality of our work. However, we would like to note that although stress in general and drug-related stress has indeed been frequently proposed to stimulate gametocyte production in *P. falciparum*, there is a paucity of studies providing unambiguous evidence for a specific effect of drugs on sexual conversion rates (as discussed in our Introduction and Discussion section). Thus, whether or not drugs can induce sexual conversion remains a highly controversial subject that we hope to settle with our approach and the data presented in this manuscript.

Essential revisions:1) Effect on ring stagesA main novel finding here is that ring stages of Pf are refractory to sexual conversion following drug and metabolic changes. DHA arrests ring stages before killing them, so DHA-treated rings might require a longer latent period before conversion to gametocyte. The authors should measure conversion at least 24-48hours after their measurement for DHA-treated trophozoites. In Figure 2, Did the authors test other concentrations of DHA on rings? Since they show that the DHA effect at the trophozoite stage is dose-dependent, one may expect that DHA affects the sexual conversion at the ring stage with concentrations higher than 10 nM.

We performed a new set of experiments in which parasites were exposed to different concentrations of DHA at the ring stage and sexual conversion was measured at a later time, as suggested by the reviewers. We measured gametocytemia by flow cytometry at ~24 and ~48 hpi of the next multiplication cycle. These experiments did not reveal any difference between sexual conversion rates determined at the two time points (new Figure 3—figure supplement 3), indicating that the reduction in sexual conversion after DHA exposure at the ring stage is not attributable to a slowdown of sexual development or conversion in DHA-treated cultures. This is now discussed in the text (subsection “DHA exposure at the ring stage does not enhance sexual conversion”). We also tested the effect of exposing parasites at the ring stage to higher DHA concentrations, as suggested by the reviewers, and found that sexual conversion rates progressively decrease with higher DHA concentration (data included in the new Figure 3—figure supplement 3, and described in subsection “DHA exposure at the ring stage does not enhance sexual conversion”).

2) Assessment of sexual conversion.It is not clear how and when the total % of gametocytes was measured following drug treatments (data presented in Figure 1E and Figure 2E) and whether DHA increases both the proportion of early gametocyte and subsequent development into mature gametocyte after 10-14 days or if this increase was driven merely by a reduction in asexual forms or increased RBC lysis following drug exposure. In this study, and in the previous article by Portugaliza et al., (2019), the authors define the sexual conversion rate as the proportion of parasites that convert into sexual forms at each replicative cycle, which can be estimated as the gametocytemia divided by the initial parasitemia. This is indeed how the authors calculated the sexual conversion rate when using the NF54-10.3-Tom line (measured by dividing the gametocytemia on day 3 (D3) by the initial ring stage parasitemia on D0). However, when using the NF54-gexp02-Tom reporter line, the authors used another calculation (subsection “Determination of sexual conversion rates”) ("the sexual conversion rate was calculated as the sexual stage parasitemia divided by the total (sexual + asexual) parasitemia, and expressed as percentage"). This corresponds to the proportion of gametocytes and not exactly to the sexual conversion rate, which is over-estimated when the number of asexual parasites is reduced due to the effects of the drug. Therefore, the axis label in graphs 1D, 2D, 3B, 4B, 5C, etc…should be changed to something like "proportion of gametocytes". The authors should consider using a more accurate calculation of the sexual conversion, as they did in their 2019 paper with the NF54-gexp02-Tom parasites or in this study with the NF54-10.3-Tom.

We agree with the reviewers that correct determination of sexual conversion rates is critical for this study. From the reviewers’ comments we feel that we did not explain our methods with sufficient clarity in the first version of the manuscript. Therefore, we have now added Figure 1—figure supplement 1 in which the different methods that we used to determine sexual conversion rates are described in detail.

In all of our approaches and with all parasite lines, the determination involves calculating the proportion of Generation 1 parasites that develop into sexual forms (Generation 0 are the parasites that are exposed to the different conditions and Generation 1 are the parasites of the next cycle), as in our previous studies. Please note that the “total (asexual + sexual) parasitemia” used in experiments with the *gexp02-Tom* reporter lines is indeed equivalent to the “initial ring stage parasitemia on Day 0” used with other parasite lines (see Figure 1—figure supplement 1). Gametocytemia was measured at different times depending on the marker used, but given that gametocytes are non-replicating, this does not affect the determination of sexual conversion rates. Using the *gexp02-tdTom* line we previously showed that gametocytemia is stable between 10-15 and 96-101 hpi, such that sexual conversion rates determined by analysis at these different times were almost identical (Portugaliza et al., 2019).

Regarding the question of how and when parasitemia and gametocytemia were determined in Figure 1E and Figure 3E, they were determined by flow cytometry in Generation 1. Values in panels E are from the same flow cytometry determination used to calculate conversion rates, which was performed at the time indicated in the schematic (panel A) included in each figure. To make this more clear, we have included this information in several figure legends (Figure 1E, Figure 3E, Figure 4C, Figure 5C, Figure 6D and several Supplementary figures) and in Results sectionsubsection “Exposure to DHA at the trophozoite stage enhances sexual conversion”. Figure 1E shows that the increase in the sexual conversion rate shown in Figure 1D does not result merely from a reduction in asexual forms, but rather from a net increase in gametocytemia, as explained in the text (subsection “Exposure to DHA at the trophozoite stage enhances sexual conversion”). No erythrocyte lysis was observed in any of our experiments at any of the drug concentrations tested.

Furthermore, as part of the mosquito infection experiments (new Figure 2, see below), we measured gametocytemia on day ≥10, and found that the observed increase in sexual conversion rate and early gametocytemia in DHA-treated cultures (pulse at the trophozoite stage) results in an increase in gametocytemia measured at day ≥10 (mature gametocytes).

We hope that the new figure and the changes in the figure legends and the text make our methods to determine the sexual conversion rate more clear, and the reviewers agree that the labeling of the axis is correct in all figures.

3) A major question is whether DHA exposure induction of sexual conversion is via a choline-dependent or independent pathway. Choline depletion induces sexual conversion and may present as a confounding effect in DHA treated trophozoites since DHA alkylates lipids, especially in trophozoites. Choline measurements should have been done in the DHA or CQ treated parasites.

We fully agree with the reviewers that understanding the mechanism of stimulation of sexual conversion by DHA and the potential involvement of LysoPC/ choline in this mechanism would be of great interest, but we consider that this is beyond the scope of this manuscript. In addition, addressing this molecular mechanism would involve using metabolomics techniques that are not established technologies in our laboratories (e.g., mass spectrometry determination of the metabolites in the biochemical pathways in which choline and LysoPC are involved). This would be a full new study in itself, because to address the mechanism meaningfully these experiments should be performed under different conditions and at multiple stages. We hope that the reviewers agree that the publication of our findings, including the first comprehensive characterization of the effect of DHA on sexual conversion, should not be delayed to perform experiments on the underlying mechanisms, which should be the subject of future investigations.

However, we have performed a new set of experiments in which we found that a heat shock at the trophozoite stage also enhances sexual conversion. While we are aware that these experiments do not directly address the mechanism of induction of sexual conversion by DHA, these observations suggest that sexual conversion can be stimulated by types of stress that are not known or predicted to have an impact on choline levels. We have included these new results in the manuscript (new Figure 7, described and discussed in the Discussion section).

4) It seems that only one single transmission experiment was performed (Figure S5), with a very small number of mosquitoes. The authors should repeat this experiment to strengthen the conclusion that ART enhances production of functional gametocytes.

We performed new mosquito infection experiments that confirmed the idea that gametocytes produced from DHA-treated cultures are fully viable and infectious. In these experiments we observed an increase in the number of mature gametocytes and exflagellating centers in cultures treated with 5 nM DHA, following the increase in sexual conversion (relative to control cultures). In addition, when mosquitoes were fed with these cultures, an increase in the number of oocysts per mosquito and sporozoites per mosquito (relative to control cultures) was observed. The data from these additional experiments strongly suggests that exposure to low concentration DHA at the trophozoite stage enhances not only sexual conversion but also transmission to mosquitoes. The new results are presented in the new main Figure 2 and described in subsection “Exposure to DHA at the trophozoite stage enhances sexual conversion”.